# Using Group Model Building to Describe the System Driving Unhealthy Eating and Identify Intervention Points: A Participatory, Stakeholder Engagement Approach in the Caribbean

**DOI:** 10.3390/nu12020384

**Published:** 2020-01-31

**Authors:** Leonor Guariguata, Etiënne AJA Rouwette, Madhuvanti M Murphy, Arlette Saint Ville, Leith L Dunn, Gordon M Hickey, Waneisha Jones, T Alafia Samuels, Nigel Unwin

**Affiliations:** 1George Alleyne Chronic Disease Research Centre, University of the West Indies, “Avalon”, Jemmott’s Lane, Bridgetown BB11115, Barbados; madhuvanti.murphy@cavehill.uwi.edu (M.M.M.); wmkjones@hotmail.com (W.J.); alafia.samuels@cavehill.uwi.edu (T.A.S.); nigel.unwin@mrc-epid.cam.ac.uk (N.U.); 2Nijmegen School of Management, Radboud University, Heyendaalseweg 141, 6525 AJ Nijmegen, The Netherlands; e.rouwette@fm.ru.nl; 3Department of Natural Resource Sciences, McGill University, Macdonald-Steward Building, 21111 Lakeshore Road, Sainte-Anne-de-Bellevue, QC H9X 3V9, Canada; arlette.saintville@mail.mcgill.ca (A.S.V.); gordon.hickey@mcgill.ca (G.M.H.); 4Institute of Gender and Development Studies, University of the West Indies, Mona Campus, Kingston 7, Jamaica; leith.dunn@uwimona.edu.jm; 5Caribbean Institute for Health Research, University of the West Indies, Mona Campus, Kingston 7, Jamaica; 6MRC Epidemiology Unit, University of Cambridge, Level 3 Institute of Metabolic Science, Addenbrooke’s Treatment Centre, Cambridge CB2 0SL, UK; 7European Centre for Environment and Human Health, University of Exeter Medical School, Knowledge Spa, Royal Cornwall Hospital, Truro, Cornwall TR1 3HD, UK

**Keywords:** NCDs, unhealthy diet, Caribbean, small islands, food sovereignty, systems science, group model building, agricultural development

## Abstract

Many Small Island Developing States of the Caribbean experience a triple burden of malnutrition with high rates of obesity, undernutrition in children, and iron deficiency anemia in women of reproductive age, driven by an inadequate, unhealthy diet. This study aimed to map the complex dynamic systems driving unhealthy eating and to identify potential points for intervention in three dissimilar countries. Stakeholders from across the food system in Jamaica (n = 16), St. Kitts and Nevis (n = 19), and St. Vincent and the Grenadines (n = 6) engaged with researchers in two group model building (GMB) workshops in 2018. Participants described and mapped the system driving unhealthy eating, identified points of intervention, and created a prioritized list of intervention strategies. Stakeholders were also interviewed before and after the workshops to provide their perspectives on the utility of this approach. Stakeholders described similar underlying systems driving unhealthy eating across the three countries, with a series of dominant feedback loops identified at multiple levels. Participants emphasized the importance of the relative availability and price of unhealthy foods, shifting cultural norms on eating, and aggressive advertising from the food industry as dominant drivers. They saw opportunities for governments to better regulate advertising, disincentivize unhealthy food options, and bolster the local agricultural sector to promote food sovereignty. They also identified the need for better coordinated policy making across multiple sectors at national and regional levels to deliver more integrated approaches to improving nutrition. GMB proved to be an effective tool for engaging a highly diverse group of stakeholders in better collective understanding of a complex problem and potential interventions.

## 1. Introduction

One in five members of the United Nations are Small Island Developing States (SIDS). These include 38 countries, of which 16 are in the Caribbean, with a combined population of around 42 million [1]. These countries share some social, economic, and environmental vulnerabilities, especially related to inadequate and unhealthy diets driving high burdens of two or more types of malnutrition [2]. Many of the SIDS of the Caribbean are faced with a triple burden of high and increasing rates of overweight and obesity (typically ≥30%), persistent underweight and stunting in children (≥7%), and micronutrient deficiencies including iron deficiency in women of reproductive age (≥20%) [3].

The drivers of these burdens are complex and interrelated and in the Caribbean include an unhealthy diet characterized by low consumption of fruits, vegetables, nuts, and whole grains [4,5], and high consumption of fats, refined sugars, and salt [3]. The SIDS of the Caribbean are often classified as being food insecure with a high proportion of the population identified as lacking access to safe, nutritious food to maintain a healthy and active lifestyle, and to varying extents relying on high levels of food imports [6]. The majority of these imports are highly processed, energy-dense, nutrient-poor foods which are partly responsible for the rising obesity epidemic and micronutrient deficiencies [6]. Food importation trends have shifted dramatically in the region over the past three decades. In the 1990s less than 45% of food in the region was imported, whereas by 2011 it was more than 60% in the Caribbean and as high as 95% for some countries [7]. These factors together with changing lifestyles are contributing to a high and increasing burden of noncommunicable disease (NCDs) in the region with one of the highest proportions of premature mortality in the world [8].

The United Nations (UN) Sustainable Development Goals made food security and nutrition an overall priority for the 2030 agenda (SDG 2) as a way to ending hunger, achieving food security, and improving nutrition through the promotion of sustainable agriculture [9]. In recognition of this goal, the SIDS governments adopted a set of agreed priorities for the 2030 Agenda in the SIDS Accelerated Modalities of Action (SAMOA) Pathway [10]. The Pathway supports efforts to end malnutrition in all its forms, reverse the spread and severity of NCDs, promote sustainable agricultural practices, and a number of measures to support economic development that optimizes and respects food security, resilience, and ecological stability [10]. The Global Action Programme on Food Security and Nutrition in the SIDS is the result of consultations led by the Food and Agriculture Organization of the United Nations (FAO) and in collaboration with the United Nations Department of Economic and Social Affairs (UNDESA), and United Nations Office of the High Representative for the Least Developed Countries, Landlocked Developing Countries and Small Island Developing States (UN-OHRLLS) to “strengthen cooperation and enhance integration of existing processes and strategies towards…commitments for food security, nutrition and sustainable development.” [11].

Despite regional efforts to tackle food insecurity and the growing NCDs epidemic and its determinants, progress within the Caribbean Community (CARICOM) has been limited [12,13]. Both the prevention of obesity [14] and improvements to the food system [15] will require systemic approaches that transcend disciplines and sectors. One important aspect of systems thinking is to engage with stakeholders and experts from across the system and move away from linear thinking [16]. Systems thinking is increasingly being incorporated in the development of models for understanding and intervening in the systems driving complex health problems like obesity and other NCDs [17,18] and have also been applied in analyzing food systems and food security. A systematic review of systems thinking in public health found that while the use of these methods is on the rise, opportunities exist for expanding the use of systems thinking and, in particular, in engaging with stakeholders, which the authors noted was especially promising [19,20].

System dynamics modeling (SDM), an established approach to systems thinking, engages stakeholders using group model building, a participatory approach that creates causal loop diagrams or simulation models to help analyze a complex problem and potential interventions [21]. It is designed to elicit key parameters acting together in a system to describe the core problem. Stakeholders, in partnership with researchers and modelers, connect parameters in a series of feedbacks and causal pathways, taking into account the unique experience and knowledge that the stakeholders have from within the system, resulting in diagrams that help explain the nature of a complex problem. Using the diagram, stakeholders can then visualize and identify leverage points for possible intervention and better understand the possible consequences (both beneficial and detrimental) to the core problem. Ideally the model helps to prevent or limit the severity of unintended consequences. A similar SDM approach has previously been applied in Fiji together with policymakers to elucidate and prioritize interventions for food-related policy, which informed our approach [22].

The “Improving household nutrition security and public health in the CARICOM” project (Food and Nutrition - FaN) seeks to develop gender-sensitive, stakeholder-informed interventions in the food system to help slow and reverse the trends of unhealthy eating and food insecurity, which can lead to obesity or malnutrition among males and females of different ages and backgrounds. To do this, the project used the group model building participatory approach with stakeholders to describe the system driving unhealthy eating in three project countries (Jamaica, St. Kitts and Nevis, and St. Vincent and the Grenadines) as well as to help identify potential project interventions. Stakeholders were selected from across the food system value chain including the community, industry, government, nongovernment, intergovernmental, and regional organizations.

Despite high-level calls for applying systems thinking to NCDs and obesity [14,23], systems methods were only relatively recently applied to explain complex problems in health [19,20]. The problem of unhealthy diet lends itself to systems thinking as multiple interacting sectors and various levels all contribute to creating an obesity-promoting environment [14,24]. In the Caribbean, a systems thinking approach to NCDs, including a group model building workshop, had been used with positive experiences for both researchers and participants [25] and so a similar approach was selected for this study. This included examining gender as a crosscutting issue to understand the system of food supply and demand.

In what follows, we describe the system driving unhealthy eating in these three Caribbean countries as mapped by stakeholders during the workshops. We then compare the potential leverage points that were identified, their perceived levels of priority, and how the system as mapped may be affected by such interventions. Finally, we describe stakeholder views on the utility of the group model building process.

## 2. Materials and Methods

### 2.1. Settings

The study countries were Jamaica, St. Kitts and Nevis, and St. Vincent and the Grenadines. Each has a different food system with varying levels of agricultural capacity, and food importation [6] but share high rates of overweight and obesity, persistent stunting in children, and micronutrient deficiency, particularly iron, driven in large part by an unhealthy diet (characterized by low fruit and vegetable intake and high intake of highly processed foods and sugar-sweetened beverages) (see Table 1). The costs of these imported foods tend to be lower than those needed for a healthy diet and thus the poorest households are the most affected [26].

### 2.2. Ethical Approval

Ethical approval for the group model building workshops was obtained from the University of the West Indies, Mona, Jamaica, Ethics Committee; the University of Medicine and Health Sciences, St. Kitts and Nevis; and the Ministry of Health, Wellness, and the Environment, St. Vincent and the Grenadines with a waiver of informed consent from participants.

### 2.3. Interviews and Stakeholder Selection

A total of 70 semi-structured face-to-face interviews were conducted with stakeholders across all three countries before the workshops to gain their individual perspectives on the drivers of unhealthy eating and inform the framing and planning of the workshops. In these interviews, stakeholders emphasized the importance of building knowledge and capacity in agriculture, promoting private–public sector collaboration, developing more integrated plans to ensure the resilience and sustainability of domestic food systems, improving communication across disciplinary and organizational boundaries, and enhancing cooperation in public policy processes. These broad themes were summarized and presented to stakeholders at the start of the workshops to help ‘set the scene’ for the discussions to follow.

Stakeholders from the private sector, civil society, local communities, government, and regional agencies were invited to participate in the two workshops: 25 participated in Jamaica (16 stakeholders and 9 project staff and researchers) and 38 in St. Kitts and Nevis representing St. Kitts and Nevis and St. Vincent and the Grenadines (SKN/SVG) (25 stakeholders and 13 project staff and researchers) (see Table 2 for a further breakdown of workshop participants).

### 2.4. The Group Model Building Workshops

We were guided by the principles and strategies outlined by Vennix and Luna-Reyes et al. in structuring the workshops [21,27]. In both workshops, participants were presented with a reference mode, as a simple and easily communicated graphical representation of the core problem over time [27]. Participants discussed and agreed that unhealthy eating has been increasing over time and set the time frame for modeling from 2000 to the present. Participants were then led in a variable elicitation exercise using the nominal group technique [27] where they were asked to identify the main drivers and consequences of unhealthy eating. With this list of variables, participants were divided into two groups for the Jamaica workshop (with a mix of different types of stakeholders), and three for the SKN/SVG workshop for group model building. This was done in keeping with recommendations [21] that suggest no more than 15 people should be involved in a single group model building exercise to maximize stakeholder participation and group interaction. Each group was then tasked with developing a causal loop diagram of the drivers of unhealthy eating. Each variable taken from the elicitation exercise was added in turn to a causal loop diagram, where participants identified the direction of causality and with the help of facilitators connected feedback loops and pathways into causal maps. Maps were recorded using Vensim PLE Software [28].

The maps were then presented and discussed in plenary where they were compared across groups to identify similarities and differences. The researchers leading the workshops, separate from participants, then took the maps and created one causal diagram for each workshop emphasizing the common pathways and loops identified in each group. These were then presented back to participants on the second day for collective evaluation and critique. Participants in both workshops suggested edits (see Appendix A), with some variables that had been aggregated under other headings subsequently brought out as individual factors and new pathways drawn. After the maps had been agreed upon, participants were led through various exercises to then identify potential areas for intervention. At the end of the workshops, stakeholders were asked to complete an evaluation of the GMB session using a short, voluntary, and anonymous online survey. This was followed up with semi-structured phone interviews with 22 stakeholders, who agreed to the follow-up, six months after the workshop. The results of the interviews are in the process of being analyzed and are not presented here.

### 2.5. Identifying Places to Intervene

Following model building, participants were divided into working groups (4–5 persons) according to areas of expertise along themes identified by the facilitators and agreed to by the participants. In Jamaica, these were: Local food supply and production, food importation, drivers of consumption, and policy drivers and determinants. In the SKN/SVG workshop these were: Social and cultural drivers of unhealthy eating, strengthening agriculture, engaging with communities, and government and policy (see Figure 1).

Participant working groups were asked to reflect on the feedbacks and pathways within the causal loop diagrams and identify areas and ways to intervene. They were asked to list as many areas to intervene as they saw fit and then prioritize them. Working groups were then asked to identify key stakeholders that should be involved in order to intervene to facilitate healthy eating as well as scoring the feasibility and impact of each area to intervene. Their responses were presented back to the larger group for discussion. A final prioritized list of areas for intervention was voted on by participants at each workshop (see Appendix A). Participants evaluated the potential impact of the proposed interventions for shifting unhealthy eating using the systems map.

## 3. Results

At the end of the group model building sessions, and using a common set of starting parameters for each workshop, participants from Jamaica produced two causal loop diagrams (Appendix A
Figure A1 and Figure A2) and from SKN/SVG three diagrams (Appendix A
Figure A3, St. Kitts; Figure A4, Nevis; and Figure A5, St. Vincent and the Grenadines). The final consensus maps for the Jamaica workshop and the SKN/SVG workshop are presented in Figure 2 and Figure 3.

### 3.1. Drivers of Unhealthy Eating

The main drivers of unhealthy eating for both maps can be grouped under four major areas: The local food supply, the influence of imported foods, the drivers of consumer decisions on food purchasing and preparation, and the coordination of policy and regulation across the food supply chain. Across all of these areas, participants agreed that norms on gender roles, the influence of socio-economic disparities, and unequal power relations affected the whole system.

### 3.2. Balance of Local/Imported Food Supply

Participants emphasized the importance of ensuring a steady, healthy local food supply in improving the availability and affordability of local foods. In Jamaica, they discussed a lack of capacity for transportation (bad rural roads), storage, and processing as a barrier to improving access to local foods. This barrier was relevant for consumers and for local food providers who expressed a desire to source locally but who currently preferred imported foods because of the greater consistency of their quality and availability. In Jamaica, stakeholders noted women often have less access to high-quality land for agriculture and are also more likely to be engaged in cottage industries.

Participants in both workshops saw several threats to the local food supply including a lack of knowledge and skills from farmers on practices of sustainable agriculture, use of local seeds and products, theft of agricultural products, damages and loss of crops to pests, competition from imported agricultural products, and the effects of climate change (both through changes in weather patterns and shocks of extreme weather events like hurricanes). They emphasized the potential role of government in coordinating and promoting training for agriculture, providing infrastructure, and educating the general public on local foods.

In addition, participants in SKN/SVG (Figure 3) emphasized working with communities to grow food and work directly with farmers, particularly in schools, to create a positive perception of farming and a more direct connection to fresh food supply. Participants in both workshops recognized the importance of strengthening the local food system as a way to improve its affordability and stability, but stressed that just increasing supply would not be enough to change consumption patterns towards healthy foods. Participants in Jamaica also expressed concerns regarding what they perceived to be the overuse of pesticides by local farmers.

Retailers and processors described using imported agricultural products when the local supply was not available or considered unreliable, particularly in St. Kitts where local production is low. While these could be used to increase the overall availability and affordability of healthy foods to consumers, participants said that imported foods generally undermine local food production and food sovereignty. In both workshops, participants suggested the creation of digital platforms and networks could facilitate communication on market needs between agricultural producers and food retailers as well as provide basic tips and tools to help build skills for farmers and fisherfolk.

Participants recognized that a lack of disposable income for consumers reinforced the consumption of cheap and often heavily processed foods. This drove the importation of these types of foods into the system thereby driving up consumption of unhealthy foods. Participants were also concerned about the safety of imported foods with many reporting that the product was often close to the expiration date or that mixed labeling standards made it unclear what was in the package. Participants in both workshops described the powerlessness of the small islands to make trade changes because they felt that opportunities for reducing the influence and reach of imported foods were limited due to international trade agreements but that a regionally coordinated effort had the best chance of success.

### 3.3. Gender, Social, Cultural Drivers of Consumption

Participants described an increasing preference for unhealthy foods, which has driven a shift in gender, social, and cultural norms around eating. This was represented by a feedback loop where more people eating unhealthy foods creates an environment where those eating patterns are acceptable and encouraged, thus further driving up the number of people eating unhealthy foods. These changes in norms on eating undermine the skills and knowledge necessary for healthy food preparation. The feedback loop is strengthened by the perceived higher social status associated with consuming imported highly processed foods, described by participants in both workshops. The increase also drives profits for companies producing and retailing unhealthy foods that increase budgets for marketing and advertising, further influencing the decisions of consumers. Stakeholders also pointed to advertising from US cable television as a driving force for unhealthy foods. One possible counterbalance to these reinforcing loops is to increase public awareness of the negative effects of unhealthy eating, which could, over time, lead individuals to change their eating patterns especially through education from healthcare professionals. Stakeholders from both workshops also stated that women were more likely to be making the decisions in the household on food purchases and preparation, making them a key entry point for skills strengthening for healthy eating.

### 3.4. Community Engagement

However, participants felt that the influence of peers and the media largely outweighs that of healthcare professionals. They stressed the importance of engaging with communities to promote healthier behaviors. Finally, the affordability of unhealthy packaged foods and the convenience of consuming them, especially in shifting economic situations where more adults are working longer hours outside the home, provide a potent driver of unhealthy eating. Participants discussed whether an alternative could be provided through healthy prepared foods.

### 3.5. Government and Policy Coordination

Participants saw an important role for policymakers across various sectors including agriculture, trade, health, education, and tourism to engage in improving the food environment. They stressed the importance of policies to disincentivize unhealthy foods and promote healthy ones especially through the use of fiscal interventions. Taxes, like those on sugar-sweetened beverages (SSB), were seen as a useful option in the Caribbean where rates of SSB consumption is high, although some participants in the private food sector were resistant to the idea, citing it could affect profits and limit consumer choice. Participants in Jamaica signaled the “Jamaica Moves” campaign as a positive example of public policy. Furthermore, stakeholders called for the coordination of policies across sectors as a key measure for improving healthy eating patterns and felt that this is an area that has been overlooked from a policy perspective. They also emphasized the need for coordinating fiscal measures with public health campaigns to ensure the engagement and buy-in of the private sector and the general public. The tourism industry was seen as both a threat to food sovereignty through the introduction and promotion of unhealthy foods as well as an opportunity for engagement as a major stakeholder in the region; participants felt that industry in particular had been overlooked as a public health partner.

### 3.6. Assessing Interventions in the System

Most of the reinforcing loops identified by participants that drive unhealthy eating converge on the decisions or choices made by individuals to eat unhealthy foods. Participants noted that these nested reinforcing loops are likely the most potent drivers of unhealthy eating in the system. Thus, participants looked at ways to break the loops either through government or community level interventions. Participants created a prioritized list of places and ways to intervene to promote healthy eating in the region (Table 3).

### 3.7. Improve Knowledge and Skills for Healthy Eating

In both workshops, participants conceived of a multilevel approach to improving the knowledge and skills necessary for healthy eating, including food nutritional content information and preparation methods. These skills were seen as presently lacking, but were once taught within households, often from mother to daughter, and existed implicitly in the population before the growing influence of imported and local highly processed foods. This local knowledge appears to have been eroded as new generations prefer pre-prepared, processed foods both as a social status symbol and because of less time available for healthy food preparation. Intervening at multiple levels to increase the knowledge and skills for healthy eating could help shift decisions on eating toward healthier foods. Any decrease in unhealthy eating could reverse the effect of a feedback loop shifting cultural norms around unhealthy eating, thus further affecting the decisions of individuals. However, just improving knowledge and skills would not have a direct effect on the power of advertising and marketing from the food industry, or the relative affordability of unhealthy foods, both powerful parallel drivers of unhealthy eating. Participants stressed the importance of a coordinated and sustained effort by government and communities to implement and maintain information and training campaigns.

### 3.8. Promote Healthy Eating Environments in Schools

Participants, particularly from SKN/SVG, saw intervening in schools and school feeding programs as an accessible and promising way of improving eating habits, especially among children. Participants saw the possibility of shifting cultural norms for children and youth on unhealthy eating and envisioned the school as a focal point for the community to engage members and provide outreach to parents and caregivers. Furthermore, students accessing healthy school meals would mean fewer eating highly processed foods, and their increased sensitivities to healthy eating. Two distinct threads emerged: One that focused on the household level and the other around national policy level changes. In the latter, participants were keen to improve the quality and nutritional content of school meals, reduce the influence of the unhealthy food industry in schools through regulation and limitation of access to prepackaged foods and sugar-sweetened beverages, and restrictions on direct marketing in schools and promotion of school activities by unhealthy food vendors. Despite the potential success of these policy efforts, participants acknowledged the greatest vulnerability to any school-based initiative was the food environment outside of school and the influence of the home diet. Thus, they stressed the importance of reaching beyond the limits of the school, by engaging the community, including local food vendors, to help ensure continuity in healthy eating habits developed in schools and supported by national policy.

### 3.9. Coordinate Policy and Fiscal Interventions for Healthy Eating

Participants were most supportive of measures that included the taxing of unhealthy foods and the subsidizing of healthy ones to tip the balance of access away from highly processed foods and fast foods. In this area, participants saw the role of government and coordination of policies as critical. For instance, participants supported a tax on sugar-sweetened beverages, but stressed that, in the absence of a public health information campaign, limitations on advertising, and coordination of taxes on other unhealthy foods, such a measure would have limited to no effect. Furthermore, participants felt it important to reinvest revenue raised through taxes on unhealthy foods to improve the accessibility of healthy, locally produced foods (especially fruits and vegetables) and reduce the burden resulting from unhealthy diets. Thus, any fiscal policy would need to intervene in several places in the systems map to affect the price of foods, availability of healthy foods, effects of advertising, and limitations on the influence of the food industry. Participants also recommended earmarking gains from taxes to help fund public health interventions as a direct benefit to those most affected by unhealthy eating.

### 3.10. Reduce Access to and Promotion of Unhealthy Foods

Across the Caribbean Community, countries are affected by a high reliance on imported foods, most of which are highly processed, energy-dense, and nutritionally poor foods [6]. The influence of these foods on diets was connected by participants to unhealthy eating and attendant obesity and NCDs. Participants recognized that interventions designed to reduce the impact of imported foods will be very difficult as they were at governmental policy level involving multinational trade agreements that many felt individual Caribbean countries were ill equipped to renegotiate. Thus, imported unhealthy foods often outcompete small local food products because local producers do not have the same capacity or scale for production. These imports directly affect the relative price and accessibility of unhealthy foods, which were identified as driving up unhealthy eating. Participants also described changing social structures, particularly as a result of economic development, meaning more people are spending longer hours working outside the home, and more women, who were traditionally the homemakers and preparers of meals, have entered the workforce. This has led to a reliance on fast and pre-prepared food that is often unhealthy. In particular, working people caught in this transition are also likely to be influenced by advertisements from large multinational food corporations (particularly from the United States, South America, Canada, and the United Kingdom) promoting fast foods and highly processed packaged foods. For many in the Caribbean, consuming these branded imported foods is seen as a status symbol. Participants called for returning to local foods and customs, and promoting them across the social spectrum as more desirable than imported foods.

### 3.11. Strengthen Local Agriculture

Importation of food to the Caribbean is not limited to highly processed foods, but also to fresh fruits and vegetables that compete with the small local agricultural sector. Participants described agriculture in the Caribbean Community as being more traditionally organized across short value chains, in small, individual-level production systems where farmers, fisherfolk, and livestock producers take their goods directly to market or have individual contracts with food producers and consumers. Participants identified an opportunity for strengthening and increasing agricultural production both as a way to improve the food environment and also food sovereignty. In Jamaica, stakeholders also proposed addressing the specific needs of female farmers who are more disadvantaged compared to male farmers in access to land, financial and technical inputs, and markets.

Participants suggested strengthening the knowledge, skills, and coordination of the agricultural sector as a way to establish a more reliable, high-quality, local supply of fresh agricultural products that food producers and consumers can rely on. These included the dissemination of the latest technologies for enhanced agricultural production as well as platforms for connecting farmers and intermediaries to markets to reduce food waste and food gluts or shortages. In particular, Jamaican stakeholders stated that they struggled to maintain a stable supply of fresh fruits and vegetables because of a lack of refrigeration and storage. Participants saw a role for government in building or supporting these networks and described a possible feedback loop whereby a more organized agricultural production sector would have more power to influence government and policy. Furthermore, participants saw opportunities for more community-based interventions using agricultural producers as champions for healthy eating, teaching community members how to create backyard gardens, and working directly with school feeding programs. Participants felt that this could reduce a general negative perception of agriculture as a practice for the poor or uneducated and may, over time, further bolster local production. However, participants recognized that any intervention in the agricultural sector alone would not directly ensure more consumption of healthy foods. Rather, participants called for a coordinated approach to agriculture connected to a reduction in unhealthy foods through fiscal measures, changing the relative availability and affordability of unhealthy foods, and thus taking advantage of a reinforcing loop that depends on the decisions people make to eat healthy or unhealthy foods.

## 4. Discussion

### 4.1. Stakeholder Engagement and Perspectives on the Model Building Process

Stakeholders engaged quickly with the model building process, suggesting variables and discussing how to draw connections; this led to rich discussions and refining of the mapping process. Despite a diverse group of stakeholders, each with their own sometimes conflicting interests in the food system and unhealthy eating, the points for intervention across the system were clear and collectively agreed. Stakeholders agreed that the household was the most powerful driver of eating behavior, but stressed the higher-level policy interventions in addition to simple education campaigns. Participants themselves presented the causal maps generated in small groups back to the larger group, explaining each of the pathways and highlighting illustrative examples given by their groups. In SKN/SVG, participants took photos of the final consensus model to reference and interact with as they worked in the intervention groups. Participants in both workshops described the maps as useful visualization for a complex issue and capturing general trends, although some specific situations can get lost. Participants to the workshop in SKN/SVG, in a post-workshop survey, found the workshop productive and were positive about the experience, although some felt more time was needed for discussion. All of the stakeholders in St. Kitts that completed the follow-up survey (n = 16) found the group model building process useful to explaining unhealthy eating.

### 4.2. Principal Findings

Despite regional differences in food importation and production rates, NCD risk profiles, and economic development, stakeholders from Jamaica, St. Kitts and Nevis, and St. Vincent and the Grenadines all described a common system driving unhealthy eating including local food production, importation regulations, the influence of unhealthy food producers and marketing, and how these affect social and gender norms around unhealthy eating. Participants were able to understand and apply concepts central to systems thinking: Pathways, feedbacks, and understanding time delays. Convening a diverse group of stakeholders meant that not only the complexity could be discussed in detail, but that a coordinated plan for intervention could be designed including the perspectives of stakeholders and influencers throughout the system. In addition, stakeholders were positive in their initial evaluation of the group model building process, showing that engagement lasted beyond the workshop. This reinforces the ideas put forth by Vennix and others that group model building yields not only outputs from the process, but a greater level of engagement for the participants involved [21,29]. More work is being carried out to assess the long-term perspectives of stakeholders who participated and their impressions of group model building.

In 2004, Popkin described the shifting trends of the nutrition transition, driving obesity particularly in the developing world [30]. Participants in the workshop described a similar shift away from home-prepared meals using fresh agricultural inputs to highly processed, energy-dense foods. However, rather than just recognizing this trend, participants generated complex explanatory pathways for this trend. Causal diagrams for childhood obesity in the US [31] also found that the price and availability of unhealthy foods, coupled with knowledge and skills for healthy eating, had a direct impact on unhealthy eating. Similar to this workshop, that study found that children are greatly influenced by the eating habits in the household. Wylie-Rosett and Jhangiani postulated a similar set of nested feedback loops centered on price and availability and the marketing of unhealthy foods driving obesity for low- and middle-income countries. However, their causal map was not drawn in collaboration with stakeholders [32].

The findings from this study support those core pathways and feedbacks that have been found in other regions. Another study in Australia, using stakeholders from nongovernmental organizations, government, and academia, examined inequalities in unhealthy diet and found similar core drivers (relative price and availability, social acceptability, and knowledge and skills) [33]. The causal maps presented here go beyond some of the other causal maps reviewed that focus on obesity prevention. They included a broad view with areas for agriculture, trade, government, and regional policy, as well as societal shifts. While systems methods have been applied to issues of obesity prevention [24,31,32,34,35,36], stakeholder-driven maps for unhealthy eating are few, especially for developing countries [22,37,38].

### 4.3. Pathways and Feedbacks to Unhealthy Eating: Aligning with the Global Action Programme

The FAO Global Action Programme (FAO-GAP) on Food Security and Nutrition in the SIDS sets three objectives based on a theory of change as a guide to implementation and monitoring and evaluation [11]. Many of the elements described in the workshops align well with the FAO-GAP recommendations and follow a similar multilevel approach from community level interventions to national and regional cooperation. Under the three main objectives, we see consistency with the plan from the perspective of stakeholders at many levels, some of whom may not be exposed to these high-level political documents but whose buy-in can be essential for success in interventions. Below, we examine each of the three main objectives in turn and how the recommendations put forth by the workshop can be seen as supporting the FAO-GAP. The recommendations are recognized by the FAO-GAP authors as being interdependent and multilevel with simultaneous approaches being necessary for success.

#### 4.3.1. Objective 1: Enabling Environments for Food Security

This objective calls for the strengthening of political commitment and governance, increases in capacity and resources for nutrition-sensitive food systems, and knowledge and evidence generation that includes documentation and sharing of traditional knowledge. Participants saw coordination across sectors as key to facilitating change. This aligns with the FAO-GAP. In another study, a researcher-generated causal diagram for determinants of obesity concluded that a lack of coordination of policy was a major barrier to changing the trajectory of the obesity epidemic [39]. Participants in both workshops emphasized a need for better public policy coordination at both the national and regional levels. There is some precedent for the Caribbean Community in setting common goals and targets, particularly around NCDs with the CARICOM Heads of State Declaration in Port-of-Spain “Uniting to stop the epidemic of chronic NCDs” in 2007 [12] and, while there has been some progress made in implementing policies related to that declaration, prevention strategies are not fully developed. On evaluation, the capacity of countries to meet these targets was found to be directly proportional to their size with the exception of Haiti [13]. The FAO-GAP proposes establishing interregional trade policies and platforms, something the stakeholders also raised and supported. However, they were generally pessimistic about the bargaining power even a regional body would have to renegotiate trade agreements that favor food importation.

The FAO-GAP further calls for leaders at the local level to engage with communities and stakeholders, bringing people together for the policy-making process. In this sense, the group model building model is a validated way of engaging with multilevel, multisectoral stakeholders. The causal maps here drew from the experience of stakeholders and included a wide variety of sectors. To the knowledge of the authors, this was the first time this type of systems thinking was applied to unhealthy diet in a developing region working directly with a wide range of stakeholders along the whole of the food value chain. Proponents of group model building point to the engagement of stakeholders as a key outcome and added benefit of participatory workshops [29,40,41]. While it is possible for researchers to construct these models based on evidence reviews (and has been done repeatedly for obesity prevention), research shows that the models are more readily used and engagement is higher among stakeholders when they are directly involved in model building [29,42]. The participants to these workshops recognized the complexity of the causal diagrams but described a sense of ownership and were able to easily present the interaction of feedback loops using examples to others despite their different areas of work and expertise. Public health experts are increasingly calling for the use of systems approaches for obesity prevention [14,23], including unhealthy diet, and engaging stakeholders across different sectors in systems thinking may be a powerful tool in generating coordinated policy [35,43].

#### 4.3.2. Objective 2: Sustainable, Resilient, and Nutrition-Sensitive Food Systems That Include the Management and Use of Oceans, Freshwater Resources, and Terrestrial Sources Relying on Local Crops, Sustainable Agricultural Practices, and Efficient Value Chains

Participants felt that the Caribbean Community, composed of Small Island Developing States (SIDS), is particularly vulnerable to the complex interaction of food sovereignty and a reduced power to negotiate trade agreements that increase the impact of large multinational food producers competing with local ones. In this sense, there are many parallels between the Caribbean and other SIDS [22,44,45] that also face the added threat of climate change to its already limited agricultural production system. They supported the need for sustainable, resilient, and adapted food systems to improve not only access to nutritious foods, but strengthen local agriculture and promote food sovereignty. The FAO-GAP supports community-based initiatives such as school feeding programs, which were also one of the recommendations from the workshop. But rather than produce a set of recommendations, stakeholders were able to draw and explain complex pathways of how these recommendations act in concert so that the strengthening of the agricultural sector was seen as a necessary component for improving school feeding. Small-scale holders and producers are the backbone of agricultural production in the Caribbean [46] and traditionally the use of cooperatives and coordination groups is not common so that some of the recommendations for these types of mechanisms in the FAO-GAP would be difficult to implement in the region. Without the engagement with stakeholders, it would be unlikely that experts, especially those outside the region, from intergovernmental organizations would immediately recognize cultural barriers to implementing aspects of the plan. It is here that local stakeholder-driven approaches can be most useful.

#### 4.3.3. Objective 3: Empowered People and Communities for Food Security and Nutrition

Objective 3 of the FAO-GAP calls for nutrition-sensitive social protection programs that engage whole communities to treat malnutrition in all its forms. This includes education programs, skill-building, and structural improvements to process, store, and preserve food. It also calls for integrating gender considerations in food security programs. All of these elements were raised in both workshops, with participants in the St. Kitts and Nevis workshop being especially focused on community-based approaches. The program also calls for a sharing of experience and best practice that was already a feature of the stakeholders in the workshop. Many gave personal examples of interventions that had been tried in the community and how these could be modified, with the input of stakeholders from other sectors, to improve performance and sustainability. This interaction arises organically from the group model building process and is, in the opinion of the authors, one of the added benefits to engaging directly with stakeholders.

Thus, group model building and similar participatory strategies for systems thinking may be useful in arriving at a shared view of a system to then help envision better coordinated ways of intervening [21]. Indeed, most of the studies we reviewed that applied systems methods to obesity prevention and unhealthy eating have called for better coordination of multisector and multilevel policies and interventions.

### 4.4. Strengths and Limitations

These workshops were the first of their kind to produce stakeholder-driven systems diagrams on unhealthy eating in the Caribbean using group model building. Participation in the workshop was high, well above the recommended 15 people for group model building, which meant that the modeling facilitators had to break into smaller stakeholder groups and conduct parallel modeling sessions. This led to several maps that had to be combined and reconciled by the modelers independently from participants, which is not ideal and exposes the final map to the possibility of undue influence and bias from the modelers’ interpretation. However, when the final model was presented back to participants for review and revision, few changes were made and the group appeared satisfied that the model captured the essential pathways that had been described in the sessions. Finally, the models themselves generalize what may be driving unhealthy eating across the Caribbean Community and thus may overlook local and country-level variations that could lead to different patterns. However, participants in the workshop in St. Kitts and Nevis were separated by island grouping for convenience and in the end arrived at very similar major pathways and drivers (see Appendix A) despite being grouped by geographically distinct and jurisdictionally different islands. This suggests that there may be a more generalizable system that drives unhealthy eating across the Caribbean Community, and possibly across other small island states in other parts of the world, even though local contextual variations exist.

Causal loop diagrams, and especially those related to obesity, have been criticized for being too complex and difficult to understand [47], but Siokou et al. argue in their review of group model building that the process is an essential part of developing the maps and that, from the stakeholder perspective, the results are generally positive. Work is continuing to evaluate the long-term perspectives of stakeholders with follow-up interviews. A more systematic approach to evaluating group model building is needed and also its ability to ensure a coordinated and effective response to the problem it is trying to describe.

Limited awareness of gender and ability to systematically use gender analysis tools in the system analysis process has meant that there is more that can be done in the future to improve understanding of the causes of poor nutrition and to identify entry points to promote gender-sensitive interventions to promote healthy eating and reduce the rate of NCDs. 

## 5. Conclusions

Group model building is an effective way of bringing together multiple stakeholders to describe a complex problem, develop a shared understanding, and envision intervention strategies for changing the trajectory of unhealthy eating. Despite differences across islands, stakeholders recognized a common system driving unhealthy eating and described clear intervention points that complement and help to elaborate and contextualize those described in the FAO Global Programme. The process lays the groundwork for a group of engaged stakeholders and actors to work together at tackling a complex problem, and was used together with other types of evidence to inform the interventions taken forward by the FaN project. However, the long-term success of this group model building process will depend on the continued engagement of stakeholders and decision makers.

## Figures and Tables

**Figure 1 nutrients-12-00384-f001:**
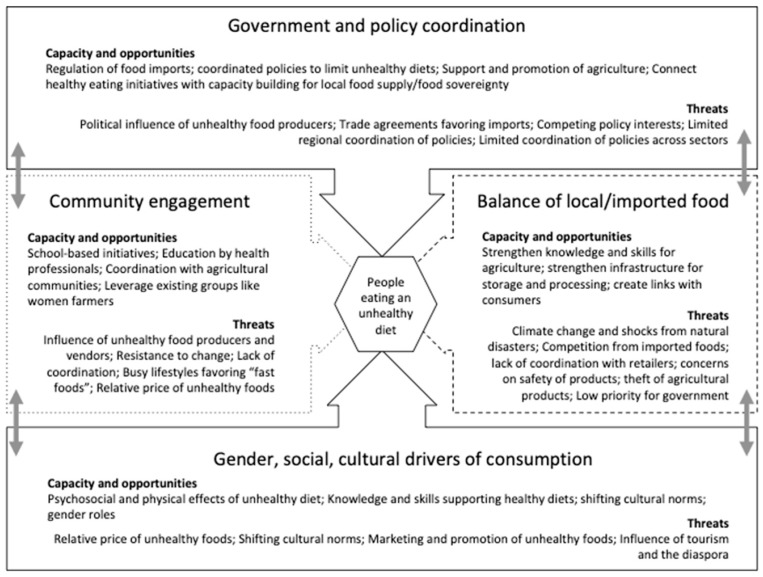
Overarching themes of drivers and places to intervene for reducing unhealthy eating.

**Figure 2 nutrients-12-00384-f002:**
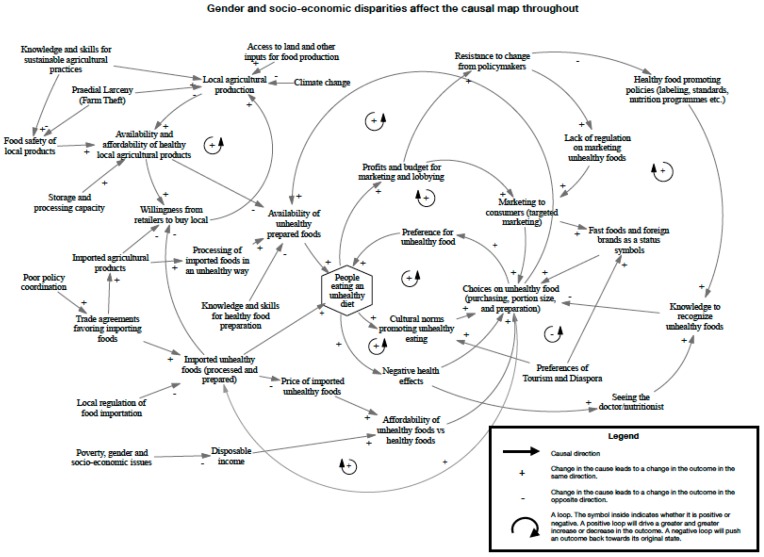
Causal map of unhealthy diet generated from group model building in Jamaica.

**Figure 3 nutrients-12-00384-f003:**
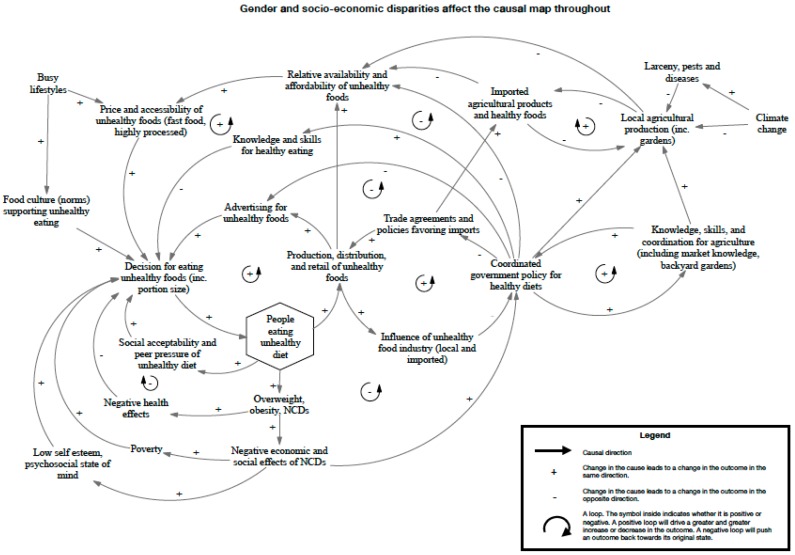
Causal map of unhealthy diet generated from group model building for St. Kitts and Nevis and St. Vincent and the Grenadines.

**Table 1 nutrients-12-00384-t001:** Summary characteristics ^1^ on aspects of malnutrition in the three study countries.

Country	Prevalence of Stunting in Children under 5 (%)	Child and Adolescent Underweight (5–19 Years) (%) (F/M)	Child and Adolescent Overweight (5–19 Years) (%) (F/M)	Adult Overweight (%) (F/M)	Anemia in Women of Reproductive Age (15–49 Years) (%)	NCD Deaths Occurring under 70 Years (%) *
Jamaica	7	14.4/14.9	30.4/29.2	63.2/47.4	22.5	31
St Kitts and Nevis	No data	16.2/17.3	28.1/27.8	59.3/45	No data	No data
St Vincent and the Grenadines	No data	16.1/16.7	29.2/28.9	61.1/48.9	24.5	44

^1^ Figures taken from the Global Nutrition Report [3] except * which is reported by the Global Health Observatory [8]. F/M – Female/Male.

**Table 2 nutrients-12-00384-t002:** Number of stakeholders across sectors participating in the group model building workshops.

Country	Government	Private Sector	Civil Society	Regional Agencies
Jamaica	Agriculture, industry and commerce *n* = 4	Restaurant *n* = 1; Food distributors *n* = 2; Export and trade *n* = 1	Vendors association *n* = 2; Market and vendors unions *n* = 3; Food production *n* = 1; Religious *n* = 2	
St Kitts and Nevis	Agriculture *n* = 1; Health and gender *n* = 6; Education *n* = 1; Foreign affairs *n* = 1	Agriculture *n* = 2	Religious *n* = 1; Agriculture and food retailers *n* = 2; civil society *n* = 2	Agriculture *n* = 2; Health *n* = 1
St Vincent and the Grenadines	Health *n* = 1; Foreign affairs and trade *n* = 1	Restaurant *n* = 1; Export and trade *n* = 1	Religious *n* = 1; Agriculture and food production *n* = 1	

*n* = number.

**Table 3 nutrients-12-00384-t003:** Prioritized list of areas for intervention, intervention leverage points, and identified key stakeholders for Jamaica and St. Kitts and Nevis and St. Vincent and the Grenadines (SKN/SVG) workshops.

Area of Intervention	Leverage Points	Key Stakeholders	Intervention Examples
Improve knowledge and skills for healthy eating	Translate research findings into policy and practice; Promote healthy eating skills in communities; Promote sustainable agriculture; Connect communities to agriculture	Agricultural sector; Education; Government; Academia; Communications and media	Training on food preparation for households; Training on sustainable best practice for agriculture; Promote local agriculture through media campaigns
Promote health eating environments in schools	Limit unhealthy foods in schools; Promote child-friendly education on healthy eating; Create networks between school community and agriculture	Community leaders; Civil society; Education	Farm-to-school schemes for including local products; Engaging food vendors on healthy options; Policies limiting unhealthy food in schools
Coordinate policy and fiscal interventions for healthy eating	Coordinate policies across sectors; Engage in public-private partnerships; Earmark taxes on unhealthy foods for promoting agriculture and treatment of NCDs	Government; Agricultural sector	Establish and maintain networks across government sectors; Creation of regional coordinating bodies
Reduce access to and promotion of unhealthy foods	Increase prices of unhealthy foods through fiscal measures; Limits on amount of imported unhealthy foods; Limit advertising of unhealthy foods	Government; civil society; Consumers; private sector	Implement fiscal measures to tax unhealthy foods and subsidize healthy local foods; Regional standards for food labeling
Strengthen local agriculture	Build technical and infrastructure capacity for agriculture; Strengthen the role of women in agriculture; Promote coordination networks from farm to market	Agricultural sector; Government; Gender specialists; Civil society	Build and maintain infrastructure to support agriculture; Use digital platforms to connect agricultural producers to food retailers

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
