# Peer review of "Using Group Model Building to Describe the System Driving Unhealthy Eating and Identify Intervention Points: A Participatory, Stakeholder Engagement Approach in the Caribbean"

_nutrients, 2020, doi:10.3390/nu12020384_

Round 1

Reviewer 1 Report

Dr. Guariguata and colleagues conducted a qualitative study based on semi-structured interviews to stakeholders in Caribbean small island developing states in order to map complex dynamic systems driving unhealthy eating and to identify potential points for intervention.

The topic is current and interesting for readers. The manuscript is well conducted and well written. I have some minor comments:

Abstract: it is not clear what "contrasting" countries means; please explain; Introduction: this section should be shortened; in particular, the authors could consider removing lines 80-91, 94-97, 105-109, 132; materials and methods:  information presented between line 148 and 157 has already contained in the introduction; the authors should eliminate repetitions.

Author Response

Abstract: it is not clear what "contrasting" countries means; please explain;

The word has been changed to "dissimilar." The countries each are different in their food importation and production practices, geography, agricultural capacity, and demography. 

Introduction: this section should be shortened; in particular, the authors could consider removing lines 80-91, 94-97, 105-109, 132;

We have removed the recommended sections in the introduction. Thank you.

materials and methods:  information presented between line 148 and 157 has already contained in the introduction; the authors should eliminate repetitions. 

The section has been shortened and simplified. Thank you.

Reviewer 2 Report

Congratulations to the authors. 

The study is of high quality, the results very interesting and necessary to understand how to address the high prevalence of obesity.

There is a part of the abstract highlighted in yellow that I suppose that it is an error. 

Author Response

Congratulations to the authors. 

The study is of high quality, the results very interesting and necessary to understand how to address the high prevalence of obesity.

There is a part of the abstract highlighted in yellow that I suppose that it is an error. 

-- We are grateful to the reviewer for the time given for comments. We did not note any section in yellow in the abstract but have reformatted the section to remove any highlighting.

Reviewer 3 Report

This manuscript describes the results intended to plot the complex dynamic system driving unhealthy eating and identify potential points for interference in three different countries. The study needs clarifications of a few concerns. 

The rationale of the study need to explain more clearly in the Introduction. Why the numbers of sketeholders given in the abstract is different from the numbers given in Materials and methods section? Legend needs to expand with self-explanatory details. Similarly, the color difference in the arrow of supplementary figures.  Feedback loop models can be differentiated with bold for better understanding.  Supplementary Figures need to be provided with high resolution.

Author Response

The rationale of the study need to explain more clearly in the Introduction. Why the numbers of sketeholders given in the abstract is different from the numbers given in Materials and methods section?

Thank you to the reviewer for the comments. The numbers for the number of stakeholders have been corrected in the abstract.

Legend needs to expand with self-explanatory details.

We have added a legend to each figure.

Similarly, the color difference in the arrow of supplementary figures.  Feedback loop models can be differentiated with bold for better understanding.  Supplementary Figures need to be provided with high resolution. 

We have edited the supplementary figures. Please note that the high resolution figures are available in a separate pdf that is being sent with the submission. While we agree that adding bold lines and colors can help make the causal maps more readable, we preferred to portray them as the stakeholders agreed to them and not make any additional changes without stakeholder approval.

Round 2

Reviewer 3 Report

The authors have carried out all the comments raised.